# Roles of Podoplanin in Malignant Progression of Tumor

**DOI:** 10.3390/cells11030575

**Published:** 2022-02-07

**Authors:** Hiroyuki Suzuki, Mika K. Kaneko, Yukinari Kato

**Affiliations:** 1Department of Molecular Pharmacology, Tohoku University Graduate School of Medicine, 2-1 Seiryo-machi, Aoba-ku, Sendai 980-8575, Japan; 2Department of Antibody Drug Development, Tohoku University Graduate School of Medicine, 2-1 Seiryo-machi, Aoba-ku, Sendai 980-8575, Japan; k.mika@med.tohoku.ac.jp

**Keywords:** podoplanin, PDPN, tumor malignancy, tumor marker, antibody therapy, cancer-specific monoclonal antibody, CasMab

## Abstract

Podoplanin (PDPN) is a cell-surface mucin-like glycoprotein that plays a critical role in tumor development and normal development of the lung, kidney, and lymphatic vascular systems. PDPN is overexpressed in several tumors and is involved in their malignancy. PDPN induces platelet aggregation through binding to platelet receptor C-type lectin-like receptor 2. Furthermore, PDPN modulates signal transductions that regulate cell proliferation, differentiation, migration, invasion, epithelial-to-mesenchymal transition, and stemness, all of which are crucial for the malignant progression of tumor. In the tumor microenvironment (TME), PDPN expression is upregulated in the tumor stroma, including cancer-associated fibroblasts (CAFs) and immune cells. CAFs play significant roles in the extracellular matrix remodeling and the development of immunosuppressive TME. Additionally, PDPN functions as a co-inhibitory molecule on T cells, indicating its involvement with immune evasion. In this review, we describe the mechanistic basis and diverse roles of PDPN in the malignant progression of tumors and discuss the possibility of the clinical application of PDPN-targeted cancer therapy, including cancer-specific monoclonal antibodies, and chimeric antigen receptor T technologies.

## 1. PDPN Structure and Functions

### 1.1. PDPN Structure

Podoplanin (PDPN)/T1α/E11 antigen/PA2.26 antigen/Aggrus possesses a heavily glycosylated N-terminal extracellular domain (approximately 130 amino acids), followed by a single transmembrane domain and a short intracellular domain (approximately 10 amino acids) (Figure 1) [1,2]. The N-terminal extracellular domain has a repeat sequence of EDxxVTPG, known as the PLAG1 to PLAG3 domains. PLAG stands for “platelet aggregation-stimulating”, which is derived from the platelet aggregation-inducing function of PDPN [3]. Furthermore, several PLAG-like domains (PLDs, one of which is also named the PLAG4 domain) with similar sequences have been identified [4,5,6,7,8,9,10,11,12]. About half of the molecular weight of PDPN comes from *O*-type sugar chains. The sugar modifications of the PDPN ectodomain are typical mucin *O*-glycans with galactose-linked β1, 3 to N-acetyl-galactosamine (GalNAc), called core 1 *O*-glycans [13,14]. Using sugar chain-deficient CHO cell lines, such as Lec1, Lec2, and Lec8, sialic acid is essential for PDPN functions [13]. The *O*-glycosylation sites at Thr52 in PLAG3 or PLD/PLAG4 have been reported to be crucial for PDPN-induced platelet aggregation [12,14].

### 1.2. Molecular Functions of PDPN Extracellular Domain

C-type lectin-like receptor 2 (CLEC-2), which is a platelet receptor, binds to PDPN and plays a critical role in PDPN-mediated platelet aggregation and lung metastases [15,16,17]. CLEC-2 was first discovered as a platelet receptor for snake venom, rhodocytin, known as a platelet aggregation-inducing toxin [18,19,20]. During platelet aggregation, both PDPN and rhodocytin can activate Src kinase families and the phospholipase Cγ2 (PLCγ2) pathway. These results suggested that PDPN is a CLEC-2 physiological ligand [15,16]. Upon PDPN binding to CLEC-2, Syk tyrosine kinase is recruited to the hemi-immunoreceptor tyrosine-based activation motif of CLEC-2. Activated Syk phosphorylates the LAT or SLP-76 adaptor proteins, which activate effector enzymes, PLCγ2, resulting in platelet aggregation [18,21,22,23] (Figure 1).

PDPN possesses well-conserved PLAG domains in the extracellular region [24] (Figure 1). In humans, the PLAG3 domain has an *O*-glycan at Thr52. Its structure is primarily a disialyl core 1 (NeuAcα2-3Galβ1-3(NeuAcα2-6)GalNAcα1-*O*-Thr) [14]. CLEC-2 was thought to recognize the sialylated PLAG3 domain since sialylation is essential for PDPN-induced platelet aggregation. However, rhodocytin can bind to CLEC-2 and aggregates platelets in the absence of carbohydrates. To elucidate the structural basis of PDPN and rhodocytin in complex with CLEC-2, the crystallographic structures were solved. Both PDPN and rhodocytin employ a two-site interaction mode. The noncanonical face of CLEC-2 is commonly used in both interactions. Other interactions are ligand-specific. Carboxyl groups from the sialic acid residues on PDPN and the C-terminus of the rhodocytin α subunit interact differently at the second binding site of CLEC-2. A similar interaction between PDPN and CLEC-2 was observed at PLD/PLAG4 domain [12].

These results indicate that CLEC-2 recognizes both sialylated *O*-glycan and the adjacent peptide of PDPN [25] (Figure 1). Furthermore, this recognition mode plays a crucial role in the strategy of tumor-specific targeting therapy (Section 5.3).

### 1.3. Molecular Functions of PDPN Intracellular Domain

PDPN is concentrated in actin-rich microvilli and plasma membrane projections, including filopodia, lamellipodia, and ruffles, where it co-localizes with ezrin, radixin, moesin (ERM) family proteins (Figure 1). Ezrin and moesin, but not radixin, can be coimmunoprecipitated together with PDPN [26]. The intracellular domain of PDPN contains juxtamembrane basic residues (RK) that act as binding sites for ERM family proteins [27]. Upon binding to them, the ezrin family proteins modulate Rho GTPase activity and promote actin cytoskeleton reorganization, which promotes cell migration, invasion, and stemness [28,29]. Interaction with ERM family proteins is essential for PDPN-mediated epithelial-to-mesenchymal transitions (EMTs) in tumor development [1,30,31], as well as lymphangiogenesis and the immune response [30]. Furthermore, two serine residues in the intracellular domain are phosphorylated by protein kinase A and cyclin-dependent kinase 5, which suppresses cell motility [31]. These results suggest that the phosphorylation of the intracellular domain may affect PDPN–ERM protein interaction and Rho GTPase activation.

## 2. Physiological Roles of PDPN

### 2.1. Lungs

PDPN is identical to T1α, which encodes an antigen expressed at the apical membrane of lung type I alveolar epithelium [32,33]. Lung type I alveolar cells cover more than 95% of the alveolar surface and are essential for gas-exchange functioning. During lung morphogenesis, the PDPN expression pattern changes from widespread in the embryonic lung epithelium to an expression that is restricted to type I alveolar cells of the distal epithelium [32,34]. Mice lacking *Pdpn* die shortly after birth due to respiratory defects. The lungs of *Pdpn KO mice* did not sufficiently fill with air. The distal lung showed marked dense cellularity and abnormal terminal respiratory units, but only a few attenuated type I cells [35,36]. These results indicate that PDPN regulates the proliferation and differentiation of lung type I alveolar epithelial cells.

### 2.2. Lymphatic Endothelial Cells

PDPN expression in the lymphatic endothelium was reported and named as “E11 antigen” [37]. PDPN is one of the most highly expressed lymphatic-specific genes; however, it is not expressed in blood vascular endothelial cells [38]. Therefore, PDPN is employed for pathological diagnosis as a highly specific lymphatic endothelial cell marker. Until the discovery of PDPN, there was no way to distinguish between blood and lymph vessels in pathological diagnosis, and specific antibodies against PDPN greatly improved the accuracy of pathological diagnosis. PROX-1, a lymphatic-specific homeobox protein, regulates PDPN in lymphatic endothelial cells [39].

*Pdpn**KO mice* also exhibited impaired lymphatic transport and lymphedema formation [35]. Furthermore, the PDPN-CLEC-2 interaction is essential for platelet aggregation and embryonic blood lymphatic vascular separation. In embryonic development, lymphatic network formation is initiated from the formation of lymph sacs, which sprout from the cardinal vein. This separation is essential for lymphatic system development [40,41]. Uhrin et al. demonstrated that PDPN-mediated platelet activation is involved in the separation process. Platelet aggregation initiates the separation at the zone of PDPN-expressing lymph sacs and cardinal veins. This phenotype was not observed in *Pdpn* KO embryos. A similar phenotype is also induced via the treatment of pregnant mice with acetyl salicylic acid, PDPN-blocking antibodies, or through inactivation of the *kindlin-3* gene required for platelet aggregation [42]. Furthermore, CLEC-2 KO mice also show a defect in blood lymphatic vessel separation [43]. Therefore, the interaction between endothelial PDPN and circulating platelets is crucial in separating the lymphatic vessels from the blood vascular system.

### 2.3. Podocyte

The term “podoplanin” is derived from its expression in kidney podocytes. Podocytes have foot processes that attach to glomerular capillaries at the glomerular basement membrane and play critical roles as slit diaphragm filtration barriers. This barrier depends on their highly differentiated postmitotic phenotype. PDPN is expressed on the apical surface of podocytes facing the luminal urinary side, and the loss of its expression is associated with foot process flattening and proteinuria, with decreased glomerular permeability in animal models [44,45]. These results indicate a crucial function of PDPN in maintaining normal podocyte morphology and glomerular homeostasis. However, the mechanism by which PDPN maintains the specific structure of foot processes remains unknown.

## 3. PDPN Overexpression in Cancer as a Diagnostic Marker

### 3.1. PDPN Overexpression in Tumors

PDPN expression has been reported in many cancers, including squamous cell carcinomas (head and neck, lung, uterine, oral, and esophageal carcinomas), malignant gliomas [46,47], mesotheliomas [48], bladder cancers [49], osteosarcoma [50], ovarian cancer [51], and testicular tumors [52]. Table 1 summarized the clinicopathological significance of PDPN overexpression and its association with poor prognosis. In contrast, PDPN expression in lung squamous cell carcinoma (SCC) is associated with a decreased incidence of lymph node metastases [53] and a better prognosis [54].

PDPN expression is observed not only in tumor cells but also in the tumor stroma, which includes CAFs. Elevated PDPN expression in CAFs from lung [55,56,57], breast [58], and pancreatic [59] tumors is correlated with tumor malignancy and poor prognosis (Table 1). PDPN-expressing CAFs promote tumor cell resistance to EGFR tyrosine kinase inhibitors [60]. Furthermore, PDPN-positive CAFs express high TGF-β [57] and PDPN-positive CAF cases display high CD204 TAMs and low CD8/FOXP3 T cells, which are associated with an immunosuppressive tumor microenvironment [61]. In Section 4.4, we discuss the roles of CAF in tumor stroma in detail.

### 3.2. Pathological Diagnosis by Specific Antibodies

As shown in Table 1, the monoclonal antibody (mAb) D2-40 has been employed for immunohistochemical staining for tumor diagnosis and tumor lymphangiogenesis [62]. D2-40 was originally established against an unidentified M2A protein derived from germ cell tumors [63]. Schacht et al. discovered that D2-40 specifically detects human PDPN [64]. The epitope of D2-40 was determined to be the PLAG1/2 domain [65].

Our groups previously established NZ-1 series (NZ-1, NZ-1.2, NZ-1.3, or NC-08) among anti-PDPN mAbs [16,65,66,67,68,69,70,71,72,73,74], which have also been employed for tumor diagnosis by means of immunohistochemistry (Table 1). NZ-1 possesses an exceptionally high affinity to a dodecapeptide (PA tag) with characteristically slow dissociation kinetics [75]. The crystal structure of the PA tag-NZ-1 complex revealed that NZ-1 recognizes a central segment of the PA tag peptide in a tight β-turn configuration [76], allowing the insertion of a PA tag into a loop structure. The PA tag system is widely used in protein detection and purification. Furthermore, the NZ-1-PA tag system efficiently promotes the crystallization of the target protein [77], implying that the NZ-1-PA tag system can be used as a crystallization chaperone to solve the target protein structure.

Moreover, NZ-1 series mAbs are used to collect circulating tumor cells (CTCs), which are useful indicators of micro-metastasis. However, the detection of rare tumor cells, contaminated in a vast majority of normal hematological cells, remains to be technically improved [78,79]. To detect CTCs effectively, a novel microfluidic system (CTC-chip) was developed using NZ-1 mAbs. Among them, an anti-PDPN antibody (clone NZ-1.2) effectively captured PDPN-high CTCs in peripheral blood from 15 of 22 malignant pleural mesothelioma (MPM) patients [72]. However, there is a limitation in that the above system can capture PDPN-high CTCs, but not PDPN-low CTCs from non-epithelioid MPM (low PDPN expression). To overcome this limitation, a novel CTC-detection chip was developed by combining the PDPN antibody with an anti-EGFR antibody (cetuximab). The cell-capture efficiency of the Cocktail-chip was improved and reached 100% in all histological MPM cell lines. Furthermore, the CTC counts were significantly associated with the clinical stage of non-epithelioid MPM [80]. These results provide a novel strategy for MPM diagnosis and could offer useful information for treating and predicting MPM patients’ prognoses.

### 3.3. The Mechanism of PDPN Overexpression in Tumors

In lymphatic endothelial cells, the transcriptional factor PROX-1 is a master regulator of *PDPN* transcription [39]. However, during the malignant progression of tumors, *PDPN* transcription has been reported to be regulated by multiple cytokines and transcriptional factors.

Hantusch et al. first reported on *PDPN* promoter analysis. They investigated about 2 kb of a 5’-flanking region of the *PDPN* gene and revealed a GC-rich region a d multiple Sp1, AP-4, and NF-1 sites. They characterized the molecular mechanism controlling basal *PDPN* transcription in human osteoblast-like MG63 (PDPN high) versus Saos-2 cells (PDPN low). An *in vitro* DNase I footprinting assay revealed multiple DnaseI-protected regions within the region bp −728 to −39, present in MG63 but not Saos-2 cells. Among these regions, two Sp1/Sp3 binding sites were identified as potential regions for *PDPN* transcriptional regulation. Overexpression of Sp1 and Sp3 independently increased promoter activity and *PDPN* transcription in Saos-2 cells. Chromatin immunoprecipitation (ChIP) analysis confirmed Sp1/Sp3 recruitment on the *PDPN* promoter. These results indicate that Sp1/Sp3 members constitutively bind to their binding sites of the *PDPN* promoter and stimulate transcription. Furthermore, they suggest the existence of additional transcription factor complexes at the upstream regions on the *PDPN* promoter [92].

The oncogenic transcription factor activator protein AP-1, composed of JUN and FOS, is essential for neoplastic transformation and malignant progression in skin carcinogenesis. Durchdewald et al. employed a mouse skin carcinogenesis model, K5-SOS-F transgenic mice (Fos^f/f^ SOS^+^), and demonstrated that *Pdpn* is a FOS target gene. In the mouse model, FOS-dependent PDPN expression was observed in mouse skin tumors induced by 12-O-tetradecanoylphorbol-13-acetate (TPA). *Pdpn* promoter activity was impaired in *Fos* KO mouse embryonic fibroblasts, which could be restored by ectopic *Fos* expression. Furthermore, the direct binding of FOS at the TPA-responsive element–like motif of *Pdpn* promoter was revealed through ChIP analysis. These results indicate the significance of the FOS-PDPN axis in neoplastic transformation and/or the malignant progression of skin tumors [93]. The involvement of AP-1 was also reported in the MG63 cell model, as described above [94]. Furthermore, SRC oncoprotein, an upstream regulator of FOS, utilized CAS to induce PDPN expression [95].

In primary human glioblastoma (GBM) and glioma cell lines, an inverse correlation between PDPN expression and PTEN levels was reported. Elevated PDPN was also observed in the subventricular zone of the brain in PTEN-deficient mice. These results indicate the involvement of PI3-kinase in PDPN expression. In human glioma cells lacking PTEN, reintroduction of wild-type PTEN, inhibition of PI3-kinase by LY294002, or inhibition of AP-1 activity by dominant-negative JUN and FOS resulted in potent downregulation of PDPN expression. These results indicated that the increased PDPN expression in human GBM is mediated by the loss of PTEN function and PI3K-AKT-AP-1 signaling pathway activation [96].

PDPN is involved in the tumorigenesis of oral SCC. Mei et al. demonstrated that the ErbB3-binding protein-1 (EBP1) can function as a transcriptional factor for promoting *PDPN* transcription during malignant progression. ChIP analysis revealed that EBP1 binds to the *PDPN* promoter surrounding the Sp1/Sp3 site. EBP1 overexpression promoted *PDPN* transcription and invasiveness. In contrast, EBP1 knockdown inhibited *PDPN* transcription, invasiveness, and tumor formation in immunodeficient mice. Therefore, EBP1 plays a key role in the upregulation of PDPN and contributes to oral tumorigenesis [97].

PDPN regulators were also reported through a pathological approach. In the pathological tumor section, PDPN-expressing tumor cells have been observed sometimes at the invasive front, where CD45-positive inflammatory cells infiltrate. Laser capture microscopy combined with gene expression profiling revealed that interferon-responsive gene expression is upregulated in PDPN-positive cells at the invasive front. Indeed, PDPN expression can be induced by interferon-γ (IFN-γ), transforming growth factor-β (TGF-β), and/or tumor necrosis factor-α (TNF-α) treatment in SCC cell lines. Furthermore, STAT1 knockdown (a signaling component of IFN-γ) in SCC cells suppressed tumor cell invasion in the subcutaneous tumor transplantation model [98]. The involvement of the TGF-β-SMAD pathway in PDPN expression has been reported in oral/pharyngeal SCCs [99] and fibrosarcoma cells [100]. TGF-β-induced PDPN expression was inhibited by SMAD4 knockdown or TGF-β type I receptor kinase inhibitor treatment. These results highlight the significance of inflammatory cytokines produced by inflammatory cells, stimulating PDPN expression at the invasive front of the tumor.

## 4. Roles of PDPN in Invasion-Metastatic Cascade

Tumor metastasis is a multistep biological process termed the invasion-metastasis cascade [101], which includes (1) cancer cell dissemination from primary sites, (2) the acquisition of migration/invasion phenotype, (3) intra/extravasation, (4) survival in circulation, and (5) adaptation and colonization in a distant organ. Furthermore, (6) the nonneoplastic stromal cells, including CAFs and tumor-infiltrating lymphocytes (TILs), also mediate these events, and these cells are termed the tumor microenvironment (TME). Recent advances provide insights into the relevance of PDPN in the multiple steps of the invasion-metastasis cascade (Figure 2).

### 4.1. Migration and Invasion

EMT is a cellular process in which epithelial cells acquire mesenchymal phenotypes (fibroblast-like morphology and cytoarchitecture, increased migratory capacity) and lose epithelial features (stable cell–cell junctions, apical-basal polarity, and interactions with basement membrane). In TME, EMT is triggered by various cytokines produced not only by tumor cells but also by stromal and immune cells. During EMT, changes occur in gene expression, including EMT transcriptional factors (EMT–TFs) and their targets, as well as epigenetic regulation, resulting in the suppression of these epithelial characteristics and the acquisition of mesenchymal characteristics [102,103]. PDPN has been referred to as “PA2.26 antigen,” and its forced expression promotes cell scattering, migration, and EMT-like morphological changes with the loss of epithelial markers (E-cadherin and keratin) and the upregulation of mesenchymal markers (N-cadherin and Vimentin) [26,95,96]. PDPN’s cytoplasmic tail binds to ezrin and/or moesin, which are members of the ERM protein family of membrane-cytoskeleton linkers and are required for RhoA activation and EMT induction [27,104].

PDPN is reported to interact with various migration- and invasion-promoting membrane proteins, including the hyaluronan receptor CD44 and matrix metalloproteinase 14 (MMP14, also known as MT1-MMP) (Figure 1). In a mouse skin carcinogenesis model, PDPN interacts with the standard isoform of CD44 (CD44s) during the progression to highly aggressive SCCs. CD44 is a highly glycosylated type I transmembrane glycoprotein that contains a significant number of variant isoforms (CD44v) due to alternative splicing. CD44 also binds to ERM proteins through its cytoplasmic tails [105]. PDPN and CD44 colocalize at cell-surface protrusions. PDPN-induced migration requires CD44 in MDCK cells, and knockdown of CD44 and PDPN in oral SCC cells affect cell spreading [106]. These results indicated that PDPN directly interacts with CD44 and modulates cell migration.

To invade the surrounding tissue, cancer cells must destroy and remodel the extracellular matrix (ECM), including both the basement membrane and the stromal ECM. PDPN was reported to stimulate MMP14 expression with oral SCC cell invasion by the activation of the small GTPase Cdc42 [107]. PDPN makes a complex with MMP14 and co-localizes at cell-surface protrusions, suggesting the potential of ECM destruction at sites [107]. In tumor cell invasion, not only tumor cells but also stromal cells express and secrete MMPs, which contributes to the destruction of the ECM for tumor invasion [108] (Figure 2). Furthermore, PDPN can stimulate TGF-β secretion in oral SCC cells, which activates surrounding fibroblasts, upregulating MMP2 and MMP14 expression [109]. We discuss the roles of PDPN in CAFs and TGF-β below (Section 4.4).

The recruitment of the glycoprotein by specialized cell-surface protrusions, called invadopodia, which recruit glycoproteins, is implicated in tumor cell invasion [110,111]. PDPN functions as a component of invadopodia in breast cancer and SCC cells [112,113]. The recruitment of PDPN to the adhesion ring of invadopodia requires binding to ERM proteins and association with lipid rafts. PDPN promotes invadopodia maturation and stabilization by activating the RhoC–Rho kinase (ROCK)–LIM kinase–cofilin signaling pathway, which stimulates the degradation of the ECM [112]. CD44 was also found to be a component of invadopodia, where it appears to interact with PDPN and recruit MMP14 [114,115]. However, the role of the PDPN–CD44 interaction in invadopodia assembly and maturation should be investigated in detail.

The diversity of motility mechanisms has been recognized in recent years. The most commonly observed type of motility in histological sections is the migration of groups of cells [116]. Analyses of the clusters indicate that the cells retain cell–cell adhesion and that there is communication from the leading edge and the trailing cells within each cluster [117,118]. Wicki et al. employed a transgenic mouse model of carcinogenesis and biopsies from cancer patients to investigate the functional contribution of PDPN to collective migration. Via the transgenic expression of PDPN in Rip1Tag2 transgenic mice (expression of simian virus 40 large and small T antigens under the control of insulin promoters; mouse model of pancreatic β cell carcinogenesis), PDPN caused an acceleration of tumor progression with a higher incidence of tumor invasion and tumor malignancy, without the formation of lymph-node or distant-organ metastasis. PDPN induces collective cell migration through filopodia formation via the downregulation of the activities of RhoA, Cdc42, and Rac in the absence of EMT. In an immunohistochemical analysis for PDPN and E-cadherin in the collective invading front of human tumors, PDPN is detectable exclusively in the outer cell layer of the invading front, whereas E-cadherin is expressed in all carcinoma cells [119].

Ameboid migration is characterized by a rounded cell morphology and the continuous formation of protruding membrane blebs, allowing the cell to squeeze through the ECM [120] and reducing the requirement for ECM proteolytic degradation [121]. In fact, MMP inhibitors are ineffective at inhibiting ameboid invasion [122]. RhoA, ROCK, and Myosin II signaling drive the rapid and high actomyosin contractility, which mediates the ameboid characteristics. PDPN is reported to enhance ameboid invasion in melanoma. PDPN expression in murine melanoma cells drives the rounded cell morphology, increasing motility and invasion. PDPN induces the phosphorylation of ERM proteins and drives cell blebbing and protrusions. These events are inhibited by ROCK inhibitor (GSK269962A). PDPN expression promotes the dedifferentiation of melanoma cells, and the loss of PDPN restores pigmentation and melanocyte differentiation. These findings support the role of PDPN as a functional biomarker for dedifferentiated and ameboid invasive melanoma, and as a promising therapeutic target [123].

### 4.2. Platelet Aggregation

PDPN was named “Aggrus,” which promotes platelet aggregation [3,13]. Tumor cell-induced platelet aggregation is thought to be significant for hematogenous metastasis. The intravasated tumor cells receive shear stress from blood flow and attacks from immune cells [124]. To overcome and survive in the circulation, PDPN induces platelet aggregation by binding to CLEC-2 on platelets, which promotes embolization and the evasion of immune cells [125,126] (Figure 2). Activated platelets also release factors, including PDGFs and TGF-β. PDGFs activate cell proliferation and survival signaling via the Ras-ERK and PI3K-AKT pathways, respectively [101]. TGF-β is a multifunctional protein, which promotes tumor cell migration/invasion and plasticity through EMT program activation [127]. These factors help tumor cells extravasate and proliferate at metastatic sites.

### 4.3. Stemness (Colonization)

Adult stem cells possess the ability to self-renew and produce differentiated progeny cells, which contribute to tissue regeneration. In the development of tumors, a small subset of tumor cells, called cancer stem cells (CSCs), are responsible for tumorigenesis, and confer resistance to treatments. These abilities are also important for metastatic colonization and tumor relapse post-treatment. Furthermore, EMT programs can promote stemness, which generates CSCs in many epithelial tissues [128]. PDPN has also been reported to be expressed in cells with tumor-initiating potential.

The PDPN-positive population exhibited clonal expansion ability and tumor formation in mice—characteristics of CSCs, which were initially observed in A431 SCC cells. Individual PDPN-expressing cells created large colonies more often than single cells, which did not express PDPN. Furthermore, PDPN-positive cells showed a higher tumor-initiating potential than PDPN-negative cells [129] (Figure 2). There is no significant difference in the cell cycle between PDPN-positive and negative cells; however, cell death was significantly lower in PDPN-positive cells. The knockdown of PDPN enhanced the cell death of PDPN-positive cells and prevented the formation of large colonies. Moreover, a ROCK inhibitor (Y-27632) suppressed the large colony formation of PDPN-positive cells, but not that of PDPN-negative cells [130]. Since PDPN possesses an ERM-binding domain and can activate the Rho-ROCK pathway, PDPN-mediated ROCK activation is thought to be important for the maintenance of CSC ability. Furthermore, in a collagen gel invasion assay, PDPN-positive A431 cells exhibited higher invasion activity in the presence of fibroblasts, suggesting that cancer stem cell functions of PDPN-positive A431 cells might be supported by the fibrogenic tumor microenvironment [131].

A hierarchical distribution of PDPN with other CSC markers, including CD44 and P63, was revealed in the pathological approach. In the immunohistochemical staining of lung SCC tissue, PDPN is mainly localized at the periphery of invading tumor nests with CD44 and P63. The distribution of the PDPN-positive cell region was more localized to the peripheral area of the tumor nests than that of CD44- and P63-positive cell regions. However, patients who had PDPN-positive tumors with a hierarchical pattern resulted in significantly better overall survival than those with PDPN-negative tumors [54]. Therefore, the roles of PDPN/CD44/P63-positive cells in tumorigenesis remain unclear.

### 4.4. Stromal Expression of PDPN and Its Roles in Tumors

The TME consists of the ECM, cytokines, and a large population of stromal cells, including CAFs, immune cells, endothelial cells, and adipocytes. Tumor stromal cells play crucial roles in constructing the TME, including its capacity to produce ECM, activate CAFs, suppress the immune system, and promote angiogenesis. Furthermore, tumor stromal cells are also the primary sources of inflammatory cytokines, including TGF-β, IFN-γ, and TNF-α, which are also known as PDPN inducers. Reciprocally, the TME exerts profound effects on tumor growth and progression. As shown in Table 1, PDPN-positive CAFs exhibited poor prognosis in lung SCC [57], as well as lung [55,132,133,134,135], breast [58], and pancreas adenocarcinomas [59]. In lung tumor cases, PDPN-positive CAFs promote tumor cell malignancy. The subcutaneous co-injection of human lung adenocarcinoma A549 cells with PDPN-positive vascular adventitial fibroblasts resulted in a high rate of tumor formation, lymph node metastasis, and lung metastasis compared with PDPN-negative fibroblasts [55]. Moreover, PDPN-positive CAFs are closely associated with the immunosuppressive tumor microenvironment. PDPN-positive CAF cases display high CD204^+^ tumor-associated macrophage (TAM) levels and a low CD8/FOXP3 T cell ratio [61]. TAMs, mainly M2 TAMs, promote the malignant progression of tumors by producing cytokines involved in angiogenesis, tumorigenesis, matrix remodeling, and immunosuppression [136]. A growing number of studies have shown that FOXP3-positive regulatory T (Treg) cells in TME secrete immunosuppressive cytokines, including IL-10 and TGF-β, and inhibit CD8-positive cytotoxic T cells, allowing tumor cells to escape the host’s immune surveillance in several cancers [137]. Furthermore, the gene expression profiles of immunosuppressive cytokines were compared using The Cancer Genome Atlas microarray lung SCC data between a PDPN-high group and a PDPN-low group. The PDPN-high group exhibited significantly higher expression of TGF-β1, interleukins (IL-1A, IL-1B, IL-6, IL-10), chemokines (CCL2), and growth factors (PDGF-A and B, FGF2) than those in the PDPN-low group. Among them, TGF-β1 expression was higher in patient-derived PDPN-positive CAFs. Immunohistochemical analyses revealed that more PDPN-positive CAFs showed higher expression of TGF-β1, which was associated with CD204^+^ TAM infiltration in lung SCC. TGF-β can inhibit the activation and proliferation of effector T cell subsets, including T_H_1, T_H_2 cells, and cytotoxic T lymphocytes. In contrast, TGF-β promotes differentiation into Treg cells from naïve CD4^+^ T cells [138]. These results indicate that PDPN-positive CAFs were associated with the immunosuppressive TME, probably due to TGF-β activation (Figure 2). However, it should be investigated whether PDPN triggers the TGF-β-mediated construction of the immunosuppressive TME. Since TGF-β crucially suppresses the immune system, TGF-β inhibitors are currently on clinical trials in advanced solid tumors combined with immune checkpoint inhibitors [139]. There is a possibility that PDPN-positive CAFs will become one of the key biomarkers for determining the immunosuppressive TME mediated by TGF-β.

### 4.5. Roles of PDPN in T Cell Immunity

The expression of co-inhibitory receptors, such as CTLA-4 and PD-1, on effector T cells play a key role in tumor immunity. Chihara et al. functionally validated PDPN as a co-inhibitory receptor expressed in both CD4^+^ and CD8^+^ T cells. PDPN participates in a larger co-inhibitory gene program that is driven by the immunoregulatory cytokine IL-27. In T cell-specific *Pdpn* conditional knockout mice, a significant delay in B16F10 tumor growth was observed. PDPN-deficient CD8^+^ TILs exhibited enhanced TNF production but no significant difference in IL-2, IFN-γ, or IL-10. The frequency of T cells with a severely exhausted phenotype was also decreased in *Pdpn* cKO mice. These results suggest that PDPN limits the survival of CD8^+^ TILs in the TME. They also identified a transcriptional factor, c-MAF, as a cooperative regulator of PDPN. This molecular circuit provides the basis of co-inhibitory receptors and their transcriptional regulation in T cells, suggesting the potential to control tumor immunity [140].

Peters et al. reported the roles of PDPN in CD4^+^ T cells. They observed that PDPN expressed effector T cells, infiltrating target tissues during autoimmune inflammation. Furthermore, mice harboring a T cell-specific deletion of *Pdpn* developed exacerbated spontaneous autoimmune encephalomyelitis with increased accumulation of effector CD4^+^ T cells in the central nervous system (CNS). In contrast, T cell-specific overexpression of PDPN exhibited defects in T cell expansion and survival. As a result, the mice exhibited the rapid remission of CNS inflammation, indicated by a reduced effector CD4^+^ T cell number in the CNS. These results also suggest that PDPN functions as an inhibitory molecule on T cells by promoting tissue tolerance and limiting the survival and maintenance of CD4^+^ effector T cells in target organs [141].

These findings strongly suggest the involvement of PDPN in immune suppression (Figure 2). However, many challenging but exciting questions remain unanswered. For example, what promotes PDPN expression in T cells? What kind of ligand(s) on tumor cells transduce the inhibitory signal to T cells? Furthermore, co-inhibitory receptors, such as immune checkpoint inhibitors, possess immunoreceptor tyrosine-based inhibitory motifs in the cytoplasmic domains [142,143]. However, PDPN does not. Recently, the neutrophil CD177 was identified as a novel PDPN receptor. Both CD177 and CLEC-2 similarly changed the PDPN-expressing CAF phosphoproteome and affected PDPN-mediated contractility [144]. Further investigations are required to elucidate the mechanism of how PDPN transduces the inhibitory signals in T cells. This is of interest, in relation to the application of this knowledge to cancer immune therapy.

## 5. Therapeutic Strategies to PDPN-Overexpressing Tumors

### 5.1. Anti-PDPN Monoclonal Antibodies (mAbs)

MAbs that target solid tumor antigens have been extensively investigated [145]. Among the FDA-approved mAbs, most mAbs receiving approval for solid tumors have targeted two members of the ERBB family, EGFR or HER2 [146,147,148]. Additionally, cytotoxic agents with conjugated mAbs against HER2 [146,147,148], Nectin-4 [149], and TROP2 [150,151] have been approved for solid tumors. A growing number of pre-clinical investigations have been reported because PDPN could be a useful diagnostic marker and an attractive molecular target for cancer therapy. Anti-PDPN mAbs have been developed and showed an antitumor effect with different mechanisms of action.

Anti-human PDPN mAb NZ-1, which recognizes the PLAG2/3 domain, has a neutralizing activity in relation to the PDPN–CLEC-2 interaction and inhibits PDPN-induced platelet aggregation and hematogenous lung metastasis [16,66] (Figure 1). Other anti-PDPN mAbs, P2-0 [152] and MS-1 [153] also recognize the PLAG2/3 domain, and 2F7 [154] recognizes the PLD/PLAG4 domain. These mAbs also suppress platelet aggregation and hematogenous pulmonary metastasis by inhibiting CLEC-2 interaction.

Non-conjugated mAbs exhibit several different mechanisms of action, including antibody-dependent cellular cytotoxicity (ADCC) and complement-dependent cytotoxicity (CDC) activities. NZ-8 is a rat-human chimeric antibody derived from NZ-1, and its ADCC and CDC activities against MPM were evaluated *in vitro* and in an *in vivo* subcutaneous tumor xenograft model [48]. NZ-8 preferentially recognized PDPN-expressing MPM but not in normal tissues. NZ-8 exhibited higher ADCC activity in the presence of human NK cells and CDC activity compared with NZ-1. Treatment with NZ-8 and human NK cells significantly inhibited tumor growth. Furthermore, chLpMab-7, a mouse-human chimeric anti-PDPN mAb with a different epitope than NZ-1, could not inhibit PDPN–CLEC-2 interaction but suppressed tumor growth and hematogenous metastasis to the lung in a neutralization-independent manner [155]. These findings suggest that ADCC and CDC activities are crucial in targeting therapeutic mAbs to PDPN-expressing tumors.

Drug-conjugated mAbs rely on direct cytotoxicity of their payloads through receptor-bound mAb–drug-conjugate endocytosis [156,157]. PMab-38, an anti-dog PDPN (dPDPN) mAb, reacts with dPDPN-expressing canine melanomas and SCCs [158,159,160]. A mouse-canine chimeric mAb (P38B) conjugated with emtansine as the payload (P38B-DM1) has been challenged for tumor therapy. P38B-DM1 showed cytotoxicity to dPDPN-expressing cells and exhibited higher antitumor activity than P38B in the xenograft model [161]. Recently, anti-PD-1 and anti-PD-L1 mAbs are used in canine tumor treatment [162,163,164]; the combination of immune checkpoint inhibitors with other antibody drugs is expected to be more effective. Anti-dPDPN mAbs may contribute to the development of canine cancer treatment, which can provide feedback for human cancer treatment. Moreover, our group has developed anti-PDPN mAbs against 18 species of animals [4,6,69,158,165,166,167,168,169,170,171,172,173,174,175,176,177,178,179]. These mAbs will contribute not only to the research on each animal but also to diagnosis and drug development. We will add novel anti-PDPN mAbs against golden hamsters and ferrets, which are small animal models of severe acute respiratory syndrome coronavirus 2 infections. Since PDPN is expressed in lung type I alveolar epithelium, these mAbs will contribute to the evaluation of the pathogenesis of virus-infected lung type I alveolar epithelial cells.

PDPN-targeted near-infrared photoimmunotherapy (NIR-PIT) has been developed. In this therapy, toxicity to tumors is induced by an antibody-conjugated photoabsorber (IR700) after exposure to NIR light [180]. NIR-PIT selectively eliminates cancer cells, resulting in local immune reactions to cancer antigens released by destroyed cancer cells. These are characterized by the rapid maturation of dendritic cells and the stimulation of cytotoxic T cells, which attack tumor cells that have evaded the initial effects of NIR-PIT [181]. An anti-human PDPN mAb NZ-1-R700 conjugate exerts an antitumor effect *in vitro* and *in vivo* in the human MPM model [73].

In a mouse xenograft and intracranial tumor model, recombinant anti-PDPN immunotoxin (NZ-1-(scdsFv)-PE38) was evaluated for treating malignant brain tumors. NZ-1-(scdsFv)-PE38 consists of the single-chain antibody variable region fragment (scFv) of NZ-1 and *Pseudomonas* exotoxin A (PE38). In a mouse xenograft model, NZ-1-(scdsFv)-PE38 exhibited high cytotoxicity to GBM and medulloblastoma cells and demonstrated a delay in tumor growth. Crucially, in the medulloblastoma intracranial tumor model, NZ-1-(scdsFv)-PE38 caused a significant increase in survival [70].

These findings reveal that anti-PDPN mAb-drug/immunotoxin conjugates exhibited significant potential as a targeting agent for PDPN-expressing tumors.

### 5.2. Radioimmunotherapy (RIT)

RIT is an internal radiation tumor therapy that transports radionuclides using high-affinity antibodies against tumor antigens [182]. In the clinic, anti-CD20 antibodies conjugated with β-emitters, Yttrium-90 (^90^Y) and Iodide-131 (^131^I), have been used in hematologic malignancies, such as non-Hodgkin’s lymphoma. The overall response rates are high, reaching 60–80%, with a complete remission rate of 15–40% [182,183]. However, the clinical efficacy of RIT for solid tumors is still low, probably due to the low radiosensitivity of solid tumors. Anti-PDPN mAb NZ-1 conjugated with ^131^I was first evaluated to be internalized into glioma cells and delivered to malignant glioma-bearing mice [67]. The ^131^I-labeled NZ-1 was efficiently internalized into LN319 GBM cells and accumulated in D2159MG GBM xenograft-bearing mice. These results showed the potential utility of NZ-1 in antibody-based therapy against GBM.

Next, the ^90^Y-labeled anti-PDPN antibody NZ-12 was reported to inhibit tumor growth in mesothelioma NCI-H226 xenograft-bearing mice. However, there was no complete remission [184]. Therefore, overcoming radioresistance is also essential for enhancing the clinical efficacy of RIT against solid tumors. Another anti-PDPN antibody, NZ-16 conjugated with Actinium-225 (^225^Ac), has been developed to improve the therapeutic effect of RIT using an anti-PDPN antibody. ^225^Ac is an α-emitter that generates four α-particles in the decay chain [185] and has a greater linear energy transfer compared with β-emitters, resulting in more potent DNA damage to cells [186]. The antitumor effects of ^225^Ac-labeled NZ-16 were compared with those of ^90^Y-labeled NZ-16 in NCI-H226 xenograft-bearing mice. RIT with ^225^Ac- and ^90^Y-labeled NZ-16 had a significantly higher antitumor effect than RIT with ^90^Y-labeled NZ-12. ^225^Ac -labeled NZ-16 induced larger areas of necrotic cell death, showed reduced tumor volumes, and prolonged survival, compared with ^90^Y-labeled NZ-16, without any adverse effects. These results strongly indicate that ^225^Ac-mediated RIT with NZ-16 is an effective and promising therapeutic option for MPM [187].

### 5.3. Cancer-Specific Anti-PDPN mAbs

A vital consideration in all tumor-targeted antibody drugs is the distribution of the target protein in normal tissues, which has significant implications for off-target effects [145]. We recently developed a cancer-specific monoclonal antibody (CasMab) method that uses flow cytometry and immunohistochemistry to select mAbs reacting with cancer cells but not with normal cells [188]. Using the CasMab method, we established anti-PDPN mAbs (LpMab-2 (mouse IgG_1_), LpMab-23 (mouse IgG_1_), and PMab-117 (rat IgM) [188,189] as well as an anti-podocalyxin mAb (PcMab-60; mouse IgM) [190]. LpMab-2 recognizes a glycopeptide of PDPN (Thr55-Leu64), which includes *O*-glycosylated Thr55 and/or *O*-glycosylated Ser56 [188] (Figure 1). LpMab-23 recognizes a naked peptide of human PDPN (Gly54–Leu64), especially Gly54, Thr55, Ser56, Glu57, Asp58, Arg59, Tyr60, and Leu64 of PDPN, and is a critical epitope of LpMab-23 [191]. PMab-117 recognizes the glycopeptide of PDPN (Ile78-Thr85), which includes *O*-glycosylated Thr85 (unpublished). A mouse-human chimeric mAb (chLpMab-2 [192] and chLpMab-23 [191]; human IgG_1_) exhibited high ADCC activity against PDPN-expressing cells and abolished tumor growth in xenograft models. chLpMab-23 and a mouse-human chimeric mAb of PMab-117 (chPMab-117; human IgG_1_) were also evaluated for toxicity using cynomolgus monkeys in a safety pharmacology test, and they revealed less toxicity (unpublished).

Another approach for cancer-specific anti-PDPN mAbs has been developed. The high-affinity 237mAb is specific for Ag104A fibrosarcoma, spontaneously developed in an aging mice [193]. The 237mAb did not react with other spontaneous tumors and cell lines [193,194]. The 237mAb was revealed to detect a glycopeptide of the PDPN extracellular domain that was produced as a result of a tumor-specific mutation in the Cosmc gene that abolished the enzyme core 1 β1,3-galactosyltransferase. This disrupts *O*-glycan core 1 synthesis, resulting in a tumor-specific glycopeptide antigen with a single Thr *O*-linked GalNAc (Tn antigen) [194]. In contrast to glycopeptide-specific antibodies in complex with simple peptides, 237mAb does not recognize a conformational epitope induced in the peptide via sugar substitution. X-ray crystallography revealed that 237mAb completely covers the carbohydrate moiety when interacting with the peptide in a shallow groove. Thus, 237mAb exhibits remarkable tumor specificity, with no physiological cross-reactivity to the unglycosylated peptide or the free glycan, making it an attractive target for immunotherapy [195].

### 5.4. Chimeric Antigen Receptor (CAR)-T

Currently, several cancer immunotherapies have been developed. Among them, T-cell-mediated immunotherapy is one of the most promising strategies [196]. Overall, T cell receptors (TCRs) or CARs confer the antigen specificity of T cells. CARs have the potential to treat a broad range of cancer patients compared with TCRs. Although several CAR molecules have been developed for hematopoietic malignancy, clinical applications for solid tumors are limited, probably due to their adverse effects. The most notable form of CAR T cell toxicity is “on-target off-tumor,” resulting from a direct attack against normal tissues that have shared expression of the targeted antigen. Therefore, it is critical to target specific antigens exclusively expressed on tumor cells [197].

GBM is the most prevalent and lethal primary malignant brain tumor in adults, with a 5-year overall survival rate of less than 10%. PDPN is overexpressed in mesenchymal GBM, which has the worst prognosis among GBM subtypes [89]. Furthermore, its therapeutic options are limited. After maximal surgical resection, the current standard care is only radiotherapy and the alkylating chemotherapeutic agent temozolomide. CAR T cells can recognize predefined tumor surface antigens independently of MHC restriction, which are often downregulated in gliomas. There is a lentiviral vector expressing a third-generation CAR, comprising an NZ-1-based scFv with CD28, 4-1BB, and CD3ζ intracellular domains. CAR-transduced peripheral blood monocytes have also been observed. The CAR T cells were found to be specific and effective against PDPN-positive GBM cells *in vitro*. Systemic injection of the CAR T cells into immunodeficient mice inhibited the growth of intracranial glioma xenografts *in vivo*. CAR T cell therapy that targets PDPN would be a promising immunotherapy for the treatment of GBM [198]. Furthermore, the use of PDPN-targeted CARs derived from a CasMab could be a promising strategy for clinical applications.

237mAb-based CAR T therapy has been developed. Unlike 237mAb, 237CAR T cells did not have exclusive specificity for cells expressing Tn-PDPN. 237CAR T cells did recognize both COSMC-mutant human and murine tumors but did not require murine PDPN expression. Recognition by 237CAR T cells was more permissive to amino acid substitutions and truncations of the Tn-glycopeptide epitope than those by 237Ab. 237CAR T cells can recognize not only Tn-PDPN, but also different Tn-glycopeptide antigens, such as Tn-TFRC, Tn-MUC1, and Tn-ZIP6 [199]. These results indicate the variations of glycopeptide recognition by full antibodies and scFv used in CAR. To overcome the above problem, a yeast display system of 237-scFv-CDR libraries was screened and isolated an affinity-matured variant with 30-fold higher affinity. The affinity-matured 237 CAR appears to have lost some activity against Tn-MUC1 however, CARs showed only modestly higher levels of activity against mouse Ag104A cells, compared with wild-type 237CARs. These results suggest that an increase in affinity does not translate to an increase in potency. Using the same libraries, 237-scFv specificity variants that reacted with both Tn-PDPN and Tn-MUC1 were isolated. The specificity variants exhibited dramatically higher activity against the human tumor lines tested [200]. Thus, structure-guided engineering and selection of the single 237-scFv scaffold allowed broader cross-reactivity to human antigens carrying aberrant Tn-glycans and mediated more efficient recognition of these cancer-associated antigens.

## 6. Concluding Remarks and Future Perspectives

In this review, we have focused on the functions of PDPN in relation to the malignant progression of tumors and the strategies of PDPN-targeting tumor therapy. First, PDPN promotes cell invasion through various cellular morphologies, including EMT-like, collective, and ameboid patterns. The mechanisms of the diversity of invasion have been identified in recent years. However, there is limited information on additional physiological partners or ligands to help understand the diverse functions of PDPN. Future research will investigate the relationship between the diversity of invasion and PDPN binding proteins. Furthermore, the mechanism of stemness acquisition and/or maintenance by PDPN should be elucidated.

Second, an increasing body of evidence suggests that PDPN-positive CAFs are also involved in tumor malignancy through TME modification. The association between PDPN-positive CAFs and a TGF-β-mediated immunosuppressive TME is regarded to be significant. However, the exact molecular mechanism remains to be determined. The persistent TGF-β signaling in the TME causes chronic immune imbalance, which could be selectively inhibited. Furthermore, PDPN functions as a co-inhibitory receptor, expressed on T cells. The elucidation of this mechanism sheds light on the molecular basis of immunosuppression by PDPN and provides a clue for cancer immunotherapy.

Third, the use of cancer-specific mAbs is a rational therapeutic strategy for minimizing adverse effects. The 237mAb specifically targets Tn-PDPN, but not other Tn antigens. It will be interesting to see whether this technique applies to human antigens (PDPN and others). In these cases, the selection of patients is also thought to be essential. We have established cancer-specific mAbs using the CasMab method. This technique is based on flow-cytometry-mediated CasMab selection. Therefore, detailed investigations to reveal their specificity are necessary for each CasMab. Furthermore, the application of cancer-specific mAbs to CAR T cells is an attractive strategy. Careful consideration and construction of CARs are required in order to maintain tumor specificity.

## Figures and Tables

**Figure 1 cells-11-00575-f001:**
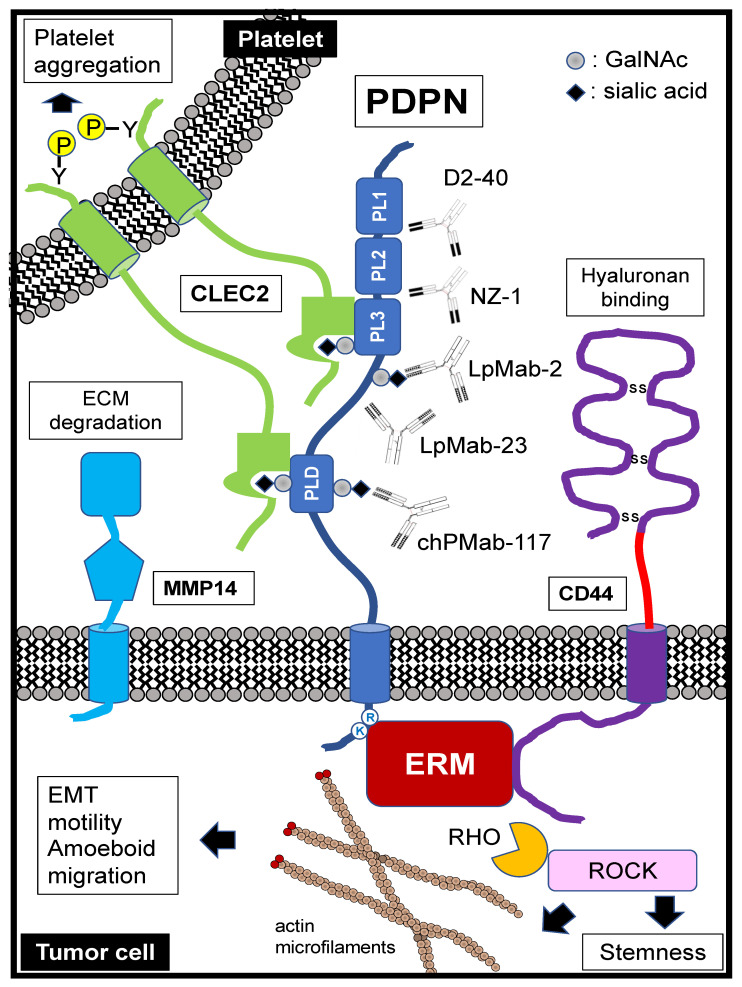
Schematic representation of podoplanin (PDPN) structure and functions. PDPN is a type I transmembrane glycoprotein consisting of an extracellular domain, a transmembrane portion, and a short cytoplasmic tail. The PDPN extracellular domain contains PLAG1-3 (PL1, PL2, and PL3) domains and PLAG-like domain (PLD). C-type lectin-like receptor 2 (CLEC-2), a platelet receptor, recognizes both the sialylated PLAG3 domain and PLD with the adjacent PDPN peptides, inducing CLEC-2 tyrosine phosphorylation and platelet aggregation. The intracellular domain of PDPN contains basic residues (RK), which function as binding sites for ezrin, radixin, and moesin (ERM) family proteins that modulate RHO GTPase activity and promote actin cytoskeleton reorganization to promote cell migration, motility, and EMT. PDPN interacts with hyaluronan receptor CD44 and matrix metalloproteinase 14 (MMP14), promoting hyaluronan-binding and extracellular matrix (ECM) degradation, respectively. Anti-PDPN mAb NZ-1 recognizes the PLAG2/3 domain, exhibits a neutralizing activity for PDPN–CLEC-2 interaction, and inhibits PDPN-induced platelet aggregation and metastasis. Anti-PDPN mAb D2-40 identifies the PLAG1/2 domain and is widely employed for immunohistochemistry. A cancer-specific mAb (CasMab) to PDPN, LpMab-2, recognizes a glycopeptide (Thr55-Leu64) of human PDPN. A CasMab to PDPN, LpMab-23 recognizes a naked peptide of human PDPN (Gly54–Leu64), especially Gly54, Thr55, Ser56, Glu57, Asp58, Arg59, Tyr60, and Leu64 of PDPN, and is a critical epitope of LpMab-23. A CasMab to PDPN, chPMab-117 recognizes the glycopeptide of PDPN (Ile78-Thr85), which includes *O*-glycosylated Thr85.

**Figure 2 cells-11-00575-f002:**
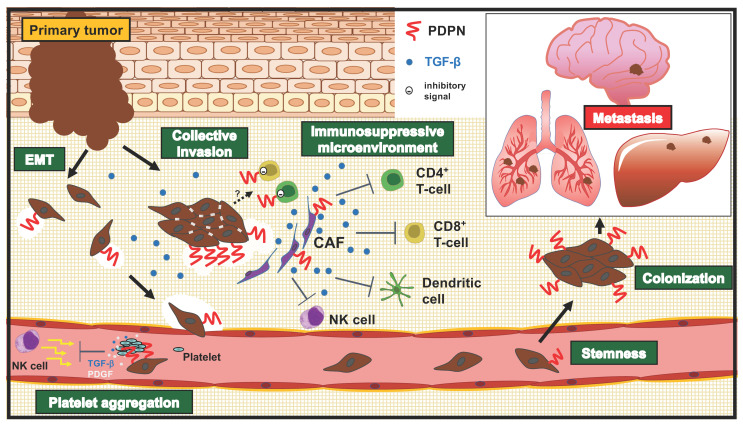
Roles of PDPN in the tumor-metastatic cascade. Tumor metastasis is a multistep biological process termed the invasion-metastasis cascade. PDPN-mediated induction of epithelial to mesenchymal transition (EMT) reduces E-cadherin levels and promotes the dissemination of cancer cells from primary sites. PDPN recruits CD44 and MMP14 at filopodia, lamellipodia, and invadopodia and stimulates hyaluronan-binding and extracellular matrix (ECM) degradation, respectively. The intracellular actin cytoskeleton reorganization by PDPN also promotes cell motility. These events confer the migration/invasion phenotype and stimulate the intravasation. Note that PDPN expression is observed at the invasive front of the tumor’s collective invasion, which implies the importance of tumor–stroma interaction. PDPN-mediated platelet aggregation promotes survival of shear stress and immune attacks in the circulation. Furthermore, platelet-derived factors (PDGFs and TGF-β) also promote the tumor cell survival and plasticity. PDPN confers stemness, probably through the Rho-ROCK pathway, and promotes colonization in a distant organ. Furthermore, PDPN-positive CAFs construct the immunosuppressive TME by producing TGF-β, a potent immunosuppressive cytokine. Furthermore, PDPN is expressed in both CD4^+^ and CD8^+^ T cells and acts as a co-inhibitory receptor, which could control tumor immunity. Thus, PDPN is involved in multiple steps of the invasion-metastasis cascade.

**Table 1 cells-11-00575-t001:** Association of PDPN expression with poor clinical outcomes.

Organ	Tumor Type	PDPN Expression	Functional and/or Clinical Significance	Detection (mAb)	Ref
Esophageal	SCC ^1^	Tumor	PDPN membrane expression is correlated with vimentin cytoplasmic expression.	IHC ^12^ (D2-40)	[81]
	SCC	Tumor	PDPN knockdown suppresses tumor formation in mice and enhances chemosensitivity.	IHC (D2-40)	[82]
	SCC	Tumor	PDPN is involved in collective cell invasion in the absence of EMT ^7^.	IHC (D2-40)	[83]
Oral	SCC	Tumor	PDPN expression correlates with cervical lymph node metastases and clinical outcome.	IHC (D2-40)	[84]
	SCC	Tumor	High PDPN expression in the biopsy specimen predicts poor response to neoadjuvant radiochemotherapy with carboplatin.	IHC (D2-40)	[85]
Skin	SCC	Tumor	High PDPN expression in the primary tumor predicts poor clinical outcomes.	IHC (D2-40)	[86]
Head & neck	SCC	Tumor	PDPN knockdown suppresses tumor migration and invasion.	IHC (NR ^14^)	[87]
Kidney	ccRCC ^2^	Tumor	High PDPN expression was an independent adverse prognostic factor for patient survival.	IHC (18H5)	[88]
Brain	GBM ^3^	Tumor	PDPN is expressed in the mesenchymal type of GBM, which presents the worst prognosis.	IHC (NZ-1.2)	[89]
Breast	AC ^4^	CAF ^6^	Tumors with a negative ER ^8^ status yielded the highest number of PDPN-expressing CAFs.	IHC (D2-40)	[58]
Lung	SCC	CAF	PDPN-positive CAFs express high TGF-β and are associated with the immunosuppressive TME ^9^.	TCGA^13^microarray	[57]
	AC	CAF	PDPN-positive vascular adventitial fibroblasts enhance tumor formation in mice.	IHC (D2-40)	[55]
	AC	CAF	PDPN-positive CAFs cases display high CD204 TAMs ^10^ and low CD8/FOXP3 T cells, associated with the immunosuppressive TME.	TCGA microarray	[61]
	AC	CAF	PDPN-positive CAFs promote tumor cell resistance to EGFR TKIs ^11^.	IHC (D2-40)	[60]
Pancreas	AC	CAF	PDPN-positive CAFs enhance the invasion of cancer cells more effectively than PDPN-negative CAFs.	IHC (D2-40)	[59]
Esophageal	AC	CAF	PDPN-expressing CAFs were observed in invasive AC, but not in precursor lesions.	IHC (D2-40)	[90]
Bile duct	CCA ^5^	CAF	Association between lymphatic vessel density and PDPN expression in CAFs. PDPN promotes the migratory ability of CAFs.	IHC (sc-134482)	[91]

^1^ SCC, squamous cell carcinoma; ^2^ ccRCC, clear cell renal cell carcinoma; ^3^ GBM, glioblastoma; ^4^ AC, adenocarcinoma; ^5^ CCA, cholangiocarcinoma; ^6^ CAF, cancer-associated fibroblast; ^7^ EMT, epithelial-to-mesenchymal transition; ^8^ ER, estrogen receptor; ^9^ TME, tumor microenvironment; ^10^ TAM, tumor-associated macrophage; ^11^ TKI, tyrosine kinase inhibitor; ^12^ IHC, immunohistochemistry; ^13^ TCGA, The Cancer Genome Atlas; ^14^ NR, not reported.

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
