# Peer review of "Roles of Podoplanin in Malignant Progression of Tumor"

_cells, 2022, doi:10.3390/cells11030575_

Round 1

Reviewer 1 Report

The review article nicely summarizes PDPN research as it stands currently. Here are some minor revisions;

  1.  It is currently not evident that serine phosphorylation indeed displaces ERM binding. The authors may modify the sentence on line 81 to "These results suggest that the phosphorylation intracellular domain may affect the PDPN-ERM proteins interactions and Rho GTPase activation."
  2.  In figure 1 it appears as though ERM is bound to CD44 not PDPN. Authors must correct this.
  3. On line 159- replace the word "Chapter" with "section".

Author Response

Dear Reviewer 1:

We are resubmitting herewith our manuscript “Roles of Podoplanin in Malignant Progression of Tumor” by Hiroyuki Suzuki et al., (cells-1576948) after carefully considering the comments raised by Reviewer 1. Here is a reply to each comment including a copy of the reviewer’s points of concern, followed by our responses. All authors in this manuscript are in agreement with all results of the revised manuscripts.

  1.  It is currently not evident that serine phosphorylation indeed displaces ERM binding. The authors may modify the sentence on line 81 to "These results suggest that the phosphorylation intracellular domain may affect the PDPN-ERM proteins interactions and Rho GTPase activation."

According to reviewer’s comment, we revised the manuscript (line 81 to 83).

      2.  In figure 1 it appears as though ERM is bound to CD44 not PDPN. Authors must correct this.

According to reviewer’s comment, we revised Figure 1.

     3. On line 159- replace the word "Chapter" with "section".

According to reviewer’s comment, we changed from "Chapter" to "section" (line 68, 161, and 328).

We appreciate all of the helpful suggestions from the reviewer. We hope the revised manuscript will meet your standards.

 Sincerely yours,

Hiroyuki Suzuki, Ph.D.

Department of Molecular Pharmacology,

Tohoku University Graduate School of Medicine 2-1, Seiryo-machi, Aoba-ku, Sendai,

Miyagi 980-8575, Japan

E-mail: hiroyuki.suzuki.b4@tohoku.ac.jp

Reviewer 2 Report

The authors reviewed the role of podoplanin in progression of malignant tumors. The roles of podoplanin in both tumor cells and interstitial cells were discussed. This would be the latest review article containing relatively new findings, and therefore valuable for relevant readers in the field. There are some comments below.

  1. There are some carcinomas in which podoplanin expression indicates good prognosis. These studies showing opposite observation should also be included in this review.
  2. It would be useful if a materials and methods section explaining study selection is provided.
  3. Making a quick research online, colorectal carcinoma, melanoma, esophagus adenocarcinoma, cholangiocarcinoma, and more are also reported that podoplanin-expressed CAF is the prognostic marker. They should be included in table 1.

Author Response

Dear Reviewer 2:

We are resubmitting herewith our manuscript “Roles of Podoplanin in Malignant Progression of Tumor” by Hiroyuki Suzuki et al., (cells-1576948) after carefully considering the comments raised by Reviewer 2. Here is a reply to each comment including a copy of reviewer’s points of concern, followed by our responses. All authors in this manuscript are in agreement with all results of the revised manuscripts.

Reviewer 2

  1. There are some carcinomas in which podoplanin expression indicates good prognosis. These studies showing opposite observation should also be included in this review.

According to reviewer’s comment, we revised the manuscript and clarified the statement about the opposite observation (line 152 to 154). In addition, we also described it (line 407 to 408).

    2.  It would be useful if a materials and methods section explaining study selection is provided.

This review article widely coveres the PDPN studies from molecular analyses to animal experiments. We feel a difficulty to explain the materials and methods in a section. We made an effort to explain a brief experimental methodology in each topic.

    3. Making a quick research online, colorectal carcinoma, melanoma, esophagus adenocarcinoma, cholangiocarcinoma, and more are also reported that podoplanin-expressed CAF is the prognostic marker. They should be included in table 1.

According to reviewer’s comment, we included esophagus adenocarcinoma and cholangiocarcinoma research in Table 1 and added the references (line 1194 to 1198).

In Table 1, we selected research articles as follows;

      i)  The prognostic data (Kaplan-Meier analysis) is included.

OR

      ii)  Both PDPN expression (histopathological or gene) analysis in clinical samples (tumor or CAF) and functional analyses are included.

In colorectal carcinoma, several papers reported that PDPN expression identified in stromal fibroblasts is a favorable prognostic marker. Other papers did not reach above criteria.

A melanoma study also did not reach above criteria.

We appreciate all of the helpful suggestions from the reviewer. We hope the revised manuscript will meet your standards.

Sincerely yours,

Hiroyuki Suzuki, Ph.D.

Department of Molecular Pharmacology,

Tohoku University Graduate School of Medicine 2-1, Seiryo-machi, Aoba-ku, Sendai,

Miyagi 980-8575, Japan

E-mail: hiroyuki.suzuki.b4@tohoku.ac.jp